behaviour

linguistics, compression, trade-offs, *Tarsius*, *Plecturocebus*, *Hylobates*

**Author for correspondence:**
Dena J. Clink
e-mail: dena.clink@cornell.edu

# Adherence to Menzerath's Law is the exception (not the rule) in three duetting primate species

## Dena J. Clink[1] and Allison R. Lau[2,3]

[1]Center for Conservation Bioacoustics, Cornell Lab of Ornithology, Cornell University, Ithaca, NY 14850, USA
[2]Animal Behavior Graduate Group, and [3]California National Primate Research Center, University of California, Davis, CA 95616, USA

  DJC, 0000-0003-0363-5581

Across diverse systems including language, music and genomes, there is a tendency for longer sequences to contain shorter constituents; this phenomenon is known as Menzerath's Law. Whether Menzerath's Law is a universal in biological systems, is the result of compression (wherein shortest possible strings represent the maximum amount of information) or emerges from an inevitable relationship between sequence and constituent length remains a topic of debate. In non-human primates, the vocalizations of geladas, male gibbons and chimpanzees exhibit patterns consistent with Menzerath's Law. Here, we use existing datasets of three duetting primate species (tarsiers, titi monkeys and gibbons) to examine the wide-scale applicability of Menzerath's Law. Primate duets provide a useful comparative model to test for the broad-scale applicability of Menzerath's Law, as they evolved independently under presumably similar selection pressures and are emitted under the same context(s) across taxa. Only four out of the eight call types we examined were consistent with Menzerath's Law. Two of these call types exhibited a negative relationship between the position of the note in the call and note duration, indicating that adherence to Menzerath's Law in these call types may be related to breathing constraints. Exceptions to Menzerath's Law occur when notes are relatively homogeneous, or when species-specific call structure leads to a deterministic decrease in note duration. We show that adherence to Menzerath's Law is the exception rather than the rule in duetting primates. It is possible that selection pressures for long-range signals that can travel effectively over large distances was stronger than that of compression in primate duets. Future studies investigating adherence to Menzerath's Law across the vocal repertoires of these species will help us better elucidate the pressures that shape both short- and long-distance acoustic signals.

# 1. Background

Identifying universal principles is a key goal in the field of animal communication. Finding commonalities across diverse taxa informs our understanding of evolutionary forces and constraints that shape diversity in communication systems [1], and may provide insight into the precursors that lead to human language [2]. In human language, whether there exist universal principles is a topic of debate, but it is generally agreed that there 'are significant recurrent patterns in organization' [3]. One of the major recurring patterns is that of compression or the minimization of redundancy in a system—an idea from information theory—wherein the efficiency of information transfer is maximized by minimizing code (or word) length [4]. The majority of human languages have been found to follow statistical patterns [5,6] that are reflections of compression [7].

Three laws, in particular, have garnered a substantial amount of interest: Zipf's Law (wherein the most commonly used word will occur approximately twice as often as the next most common word [8]), Zipf's Law of Abbreviation (elements used more frequently in a communication system are shorter [8]) and Menzerath's Law (the greater the whole, the smaller its constituents [9]). There has been increasing interest in applying these statistical laws outside of human language [10–21]. These statistical laws reflect compression, and it has been proposed that compression is a universal principle not just in vocal communication but in behaviour more broadly [1]. Importantly, finding patterns that are consistent across human language and animal communication can provide important insight into the patterns of organization of acoustic signals that were precursors to phonological and syntactic structure [22].

Adherence to Zipf's Law has been shown in dolphin whistles [10,11], gorilla gestures [23] and Carolina chickadee calls [24], but not in black-capped chickadee calls [14]. The authors attributed lack of conformity to Zipf's Law in one chickadee species (but not the other) to differences in social complexity [14]. One of the first-documented tests of Zipf's Law of Abbreviation was also in black-capped chickadees [14], and subsequent analyses showed adherence to Zipf's Law of Abbreviation in dolphin surface behaviours [25], Formosan macaque vocalizations [16], the short-range vocalizations of four bat species [17], a subset of chimpanzee gestures [12], male (but not female) rock hyrax vocalizations [18], penguin vocalizations [19], and in note [20] but not phrase usage [21] in male gibbon solos. Adherence to Zipf's Law of Abbreviation was not seen in the vocal repertoire of two New World monkeys (common marmosets and golden-backed uakaris [15]), ravens [6] or the full-body gestures of chimpanzees [12]. Altogether, studies have provided mixed results for Zipf's Law and Zipf's Law of Abbreviation.

Menzerath's Law was originally presented as a linguistic law, but has been applied to a wide range of systems outside of language, including music [26], genomes [27], genes [28] and proteins [29]. In human language, adherence to Menzerath's Law is widespread and can be found at multiple levels: longer words comprise shorter syllables, and longer sentences comprise shorter clauses [9,30]. The ubiquity of Menzerath's Law in human language is thought to reflect compression, and this has been supported mathematically [31]. Patterns consistent with Menzerath's Law in animal communication systems have been proposed to be the result of selection for efficient transfer of information [31] and/or biomechanical or energetic constraints on the production of acoustic signals [32]; note that these explanations occur at different levels of organization [33] and are not mutually exclusive. There have been few documented examples of lack of adherence to Menzerath's Law in any system and it has been proposed that adherence to the law is inevitable [34]. But genomes of gymnosperms, amphibians and ray-finned fishes do not adhere to the law [27], providing empirical evidence that adherence is not necessarily inevitable. In non-human communication systems, explicit tests of Menzerath's Law have been done in gelada vocal sequences [31], chimpanzee hoots [32] and gestures [12], penguin display songs [19] and in vocal sequences of three species of male gibbons [20,21]; all vocalizations exhibited patterns consistent with the law. A recent study on mountain gorilla close-calls found that this system only partially adhered to the predictions of Menzerath's Law [35].

There have been other cases where a negative relationship between sequence size and call duration was not found, although the authors were not explicitly testing Menzerath's Law. For example, female baboons emit grunts with a higher number of calls per bout when they are interacting with their offspring, but the mean call duration increased, a pattern opposite of that predicted by Menzerath's Law [36]. Similarly, in Barbary macaques call sequences given around the time of ovulation comprised calls of longer duration [37]. For female baboons, a lack of variation in call duration may have driven the observed pattern. For Barbary macaques, a lack of adherence to Menzerath's Law might be the result of opposing selection pressures between compression and long-range transmission [31], as

copulation calls are loud and appear to be directed at distant male receivers [37]. However, as mentioned above, subsequent findings indicated that patterns consistent with Menzerath's Law can be found in primate long-distance vocalizations [20,21,32].

Duetting evolved independently in the Primate order at least four times, and is seen in the indris (Indriidae), tarsiers (Tarsiidae), titi monkeys (Callicebinae) and gibbons (Hylobatidae [38]). Non-human primate duets tend to follow a stereotyped pattern, but there is evidence that duets emitted under different contexts (e.g. during inter-group encounters or in the presence of predators [39]) have a slightly different structure, and that conspecifics respond differently to duets emitted under different contexts [40]. In addition, certain features of primate long calls contain information about caller condition or quality. For example, male white-handed gibbons with higher androgen levels produce songs of higher frequency [41]. Younger female white-handed gibbons produce long calls with a higher maximum frequency than older females, which may be related to age-related changes in the condition of the calling animal [42]. There is also substantial evidence for individual signatures in the duet contributions of indris [43], tarsier females [44], titi monkeys [45] and gibbons [46–48]. Although the function of duets remains a topic of debate [49,50], it is clear that duets are used to transmit information about the caller(s) to conspecifics over long distances.

Here, we use existing datasets on three distantly related duetting primate species—tarsiers (*Tarsius spectrumgurskyae*), titi monkeys (*Plecturocebus cupreus*) and gibbons (*Hylobates funereus*)—to investigate whether non-human primate long-distance duet contributions adhere to Menzerath's Law. The independent evolution of long-distance vocalizations in these distantly related species is presumably the result of similar selection pressures for effective transfer of information in a forest environment, energetic constraints on calling and/or biomechanical trade-offs related to call production. There are no *a priori* reasons to predict that duets would be more (or less) likely than other parts of the vocal repertoire to adhere to Menzerath's Law. We focused on the duet contributions because these vocalizations are emitted under similar contexts across taxa, presumably shaped via similar selection pressures, and provide a useful comparative model to test for the broad-scale applicability of Menzerath's Law. Specifically, we tested for a relationship between the number of notes in a call sequence and the duration of the notes; a negative relationship is consistent with Menzerath's Law [31]. We predicted that—given the prevalence of patterns consistent with Menzerath's Law in communication systems—all calls examined would adhere to the law. To determine whether patterns consistent with Menzerath's Law in primate duets are the result of energetic or breathing constraints [31], we also assessed the importance of position in call sequence as a predictor of note duration.

# 2. Methods

## 2.1. Data acquisition and acoustic analysis

We tested for adherence to Menzerath's Law in eight different call types: tarsier female and male duet contributions [44], titi monkey pulse [51] and chirp [45] duet contributions, the introductory and trill portion of the female gibbon calls [47,52], male gibbon duet contributions (known as codas [46]) and male gibbon solos [21]. Figure 1 includes representative spectrograms for each call type included in the analysis. For all calls included in the present analysis, individual notes and calls were annotated by human observers using selection tables in Raven Pro v. 1.5 or 1.6 (Cornell Lab of Ornithology, Center for Conservation Bioacoustics, Ithaca, NY, USA). See electronic supplementary material, S1 for details on data collection and acoustic analysis for each call type. Although spectrogram settings were slightly different for each call type, we used the robust features in Raven Pro, which estimate features based on the energy of the selection; these features are more robust to inter- and intra-observer variation in note selection [54,55]. In addition, since our analysis focuses on temporal and not spectral features of the calls, slight differences in spectrogram settings (and therefore time and frequency resolution) are unlikely to have a major impact on our results.

For each call, we had the following information: individual identity, sex, call start and stop time (s), and individual note start and stop time (s). We used the Raven selection tables to calculate call duration, note duration and number of notes per call using a script written in the R programming environment [56]. We pooled male and female titi monkey duet contributions, given the previously documented lack of sex-differences in duet contributions [45,51,57]. A previous study showed that male Bornean gibbon solo phrases adhered to Menzerath's Law [21], but we include the data here as the previous

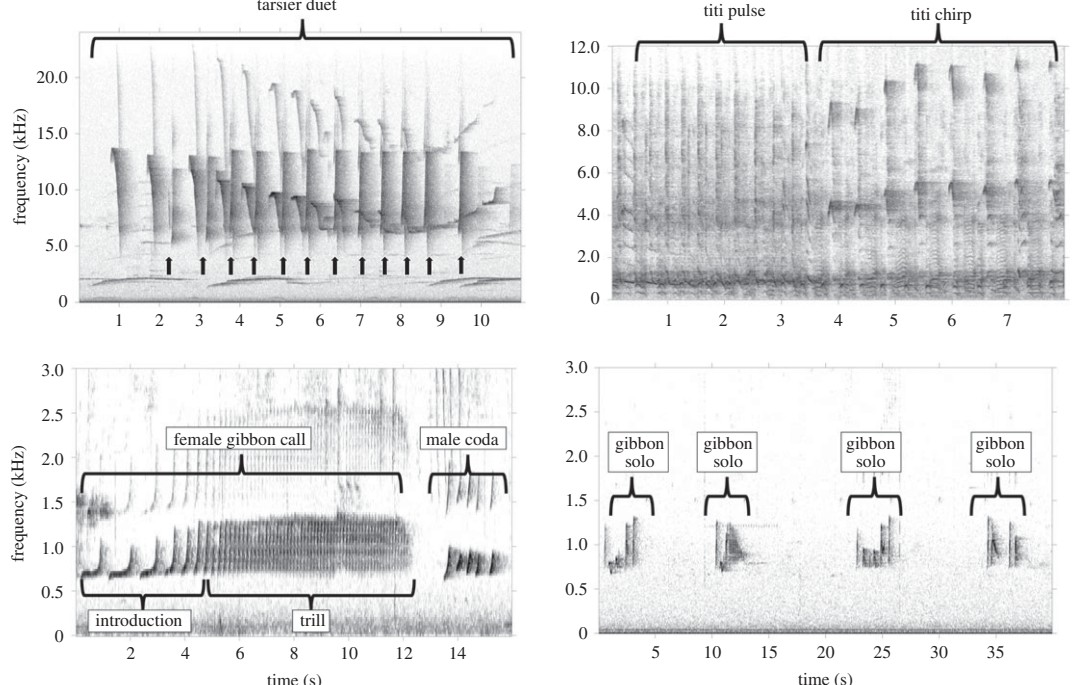

**Figure 1.** Representative spectrograms of the call types used in the present analysis. Spectrograms were made in the Matlab-based program Triton [53]. The black arrows in the tarsier spectrograms indicate male duet contributions, and the other notes are the female contribution. Note the differences in *x*- and *y*-axes of the spectrograms for enhanced visualization of the signals.

study calculated Menzerath's Law using mean note duration, and it has been suggested that patterns consistent with Menzerath's Law are inevitable if mean note duration is used [58].

## 2.2. Test of Menzerath's Law

To test for adherence to Menzerath's Law in each call type, we created four generalized linear mixed models (GLMMs) using the R package 'lme4' [59] in the R programming environment v. 3.6.2 [60]. All models included note duration (s) as the outcome. The first model included the number of notes as a predictor, the second model included the position in the call sequence as a predictor, the third model included the number of notes and position of the note in the call sequence as predictors and the fourth model (null model) included no predictors. Each model reflects a specific prediction related to the production of duet vocalizations. For example, for the first model, a negative relationship between note duration(s) and the number of notes in the call provides support for Menzerath's Law. For the second model, a negative relationship between note duration and position in call sequence indicates that duets are influenced by breathing and/or energetic constraints. The third model reflects the prediction that both Menzerath's Law and breathing or energetic constraints shaped the vocalization, and the fourth model reflects the prediction that neither of these variables are important predictors of note duration.

We analysed each call type independently, and for each of the eight call types, we created a series of four GLMMS as outlined above. Apart from the introductory notes and the trill notes of the female gibbon duet contribution, the call types included in our dataset were either general repetitions of the same note, or sufficiently graded that effective categorization by human observers into distinct note types was not possible. Therefore, for this analysis, we did not classify note types within calls. All models included individual identity as a random effect. For gibbon females, we included a random effect for site, as the dataset included calls collected from seven different sites. We compared models using Akaike information criterion adjusted for small sample sizes (AICc) implemented in the R package 'bbmle' [61]. AICc model comparison provides a relative test of model quality, but does not provide information of how well the top model(s) fit the data [62], so we also calculated a pseudo-$R^2$ value using the 'MuMIn' package [63]. The pseudo-$R^2$ value provides information about

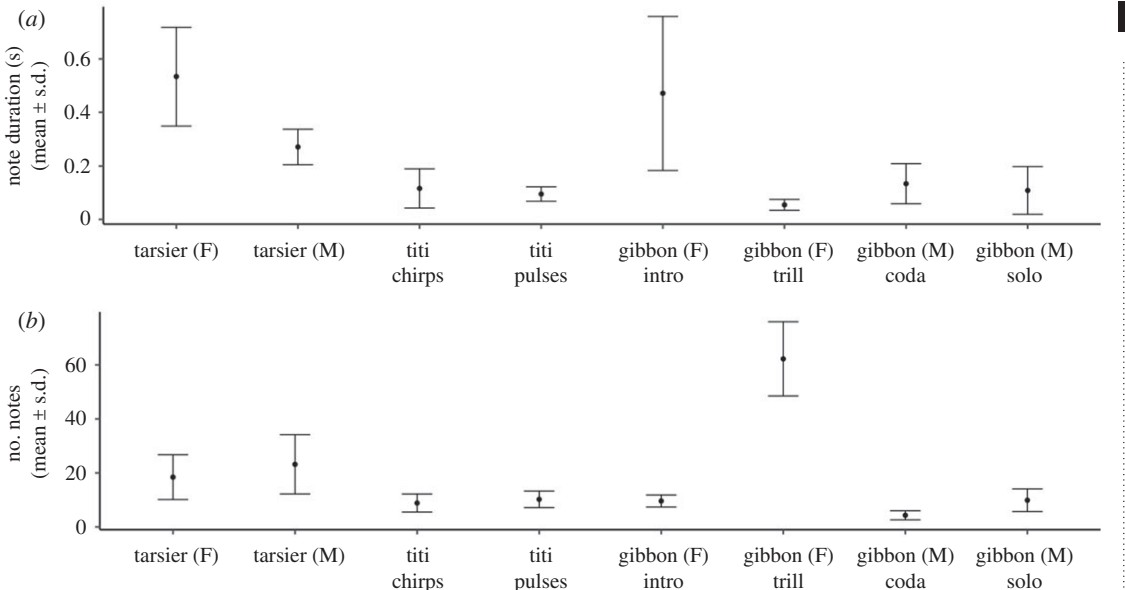

**Figure 2.** (*a*) Duration in seconds (mean ± s.d.) and (*b*) number of notes (mean ± s.d.) for call types included in the present study. See the main text and figure 1 for description of call types.

**Table 1.** Summary of total number of notes, calls and individuals for each call type included in the present analysis. We include summary statistics of the mean, minimum and maximum number of notes for each call type, along with the mean, minimum and maximum duration(s) of notes included in the analysis. The same calls were used for the gibbon female introduction and trills, so the number of individuals and the number of calls was only counted once. See the main text and figure 1 for description of call types.

| call type | total notes analysed | number of calls analysed | number of individuals | mean number of notes (range) | mean note durations (range) |
|---|---|---|---|---|---|
| tarsier (F) | 1416 | 118 | 15 | 18.43 (6–41) | 0.53 (0.18–1.77) |
| tarsier (M) | 1413 | 112 | 14 | 23.18 (6–48) | 0.27 (0.09–0.49) |
| titi chirps | 6878 | 901 | 53 | 8.85 (2–19) | 0.12 (0.02–0.55) |
| titi pulses | 3534 | 381 | 75 | 10.22 (3–19) | 0.1 (0.04–0.41) |
| gibbon (F) intro | 8968 | 1107 | 87 | 9.6 (4–22) | 0.47 (0.14–1.81) |
| gibbon (F) trill | 65 563 | ~ | ~ | 62.23 (14–113) | 0.05 (0.02–0.14) |
| gibbon (M) coda | 1798 | 482 | 35 | 4.32 (2–10) | 0.13 (0.03–0.71) |
| gibbon (M) solo | 13 836 | 2363 | 13 | 9.89 (2–27) | 0.11 (0.02–0.8) |
| total | 103 406 | 5464 | 292 | ~ | ~ |

the variance explained by the predictors (or fixed effects) of the models and the total variance explained by the models [64].

## 3. Results

We analysed eight different call types (male tarsier duet contribution, female tarsier duet contribution, titi monkey chirp, titi monkey pulse, gibbon female introduction, gibbon female trill, male gibbon coda and male gibbon solo) with a total of 103 406 notes from 5464 duet contributions from 292 individuals of three duetting primate species (see table 1 for summary of sample size by call type). There was substantial variation in note duration ranging from 0.05 s ± 0.02 standard deviation (s.d.) for gibbon female trills to 0.5 s ± 0.18 s.d. for tarsier females (figure 2*a* and table 1). The mean number of notes in calls ranged

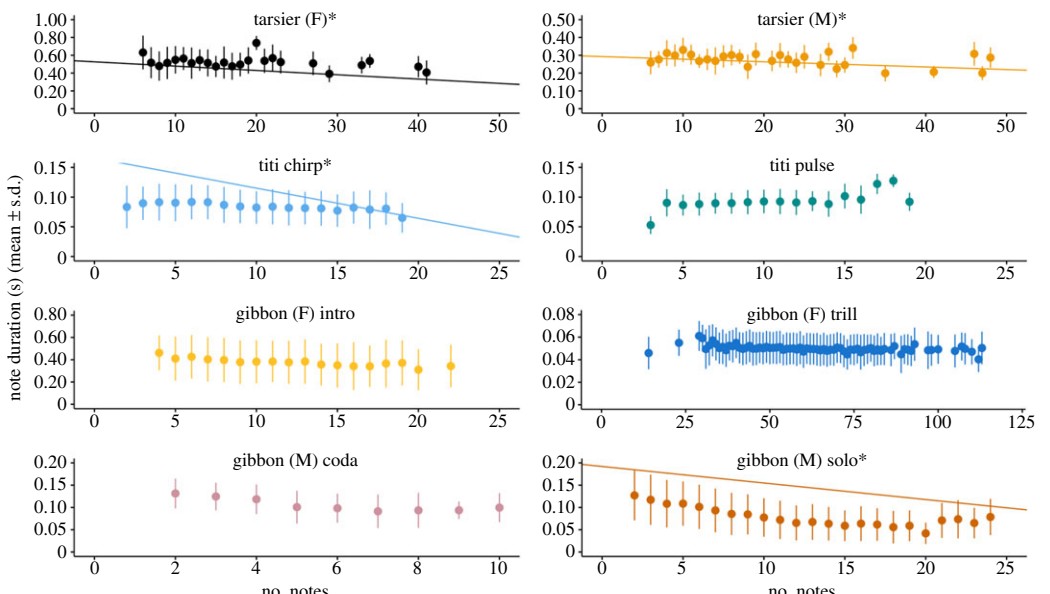

**Figure 3.** Relationship between note duration in seconds and vocal sequence size (i.e. number of notes). The points indicate the mean note duration ± s.d. The presence of regression lines as well as an asterisk next to the call type are included for call types which adhered to Menzerath's Law. Regression lines were estimated from the top model for that call type. Plots that do not have a regression line or asterisk next to the call type indicate a lack of adherence to Menzerath's Law.

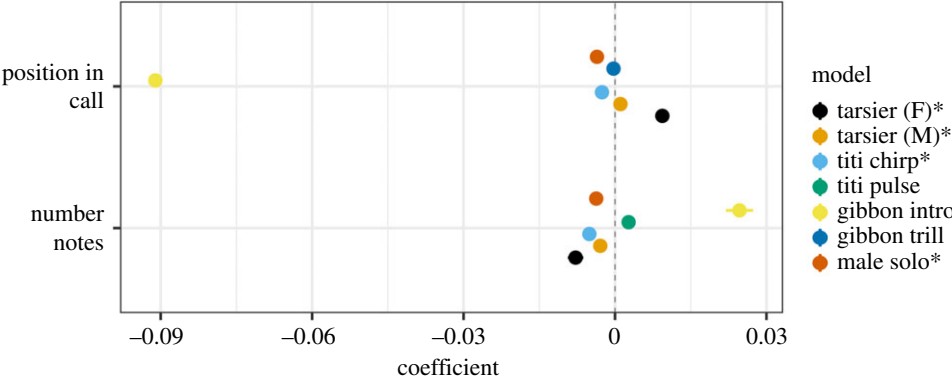

**Figure 4.** Coefficient estimates (±90% confidence intervals) for top models for call types included in the present analysis. In coefficient plots such as these, a negative coefficient estimate indicates a negative relationship between that predictor and the expected outcome (note duration), whereas a positive coefficient estimate indicates positive relationship between that predictor and expected note duration. A negative relationship between the number of notes in a call and the duration of those notes is consistent with Menzerath's Law. The model for each call type presented here was ranked highest as indicated by AICc comparison; call types that exhibited patterns consistent with Menzerath's Law are indicated with an asterisk. The top model for male gibbon codas was the null model, so the coefficient estimates for this call type were not included in the above plot.

from 4.32 ± 1.66 s.d. notes in male gibbon codas to 62.22 ± 13.74 s.d. notes in gibbon female trills (figure 2*b* and table 1).

## 3.1. Menzerath's Law in primate duets

We found patterns consistent with Menzerath's Law in four of the eight call types analysed: tarsier female, tarsier male, titi monkey chirps and male gibbon solos (figures 3 and 4). The top models for these four call types also included position in the call sequence as a predictor (figures 4 and 5). For male and female tarsiers there was a positive relationship between note duration and position in call sequence, whereas for titi monkey chirps and male gibbon solos, position in call sequence was a negative predictor of note duration (table 2; figures 4 and 5). See table 2 for a summary of the

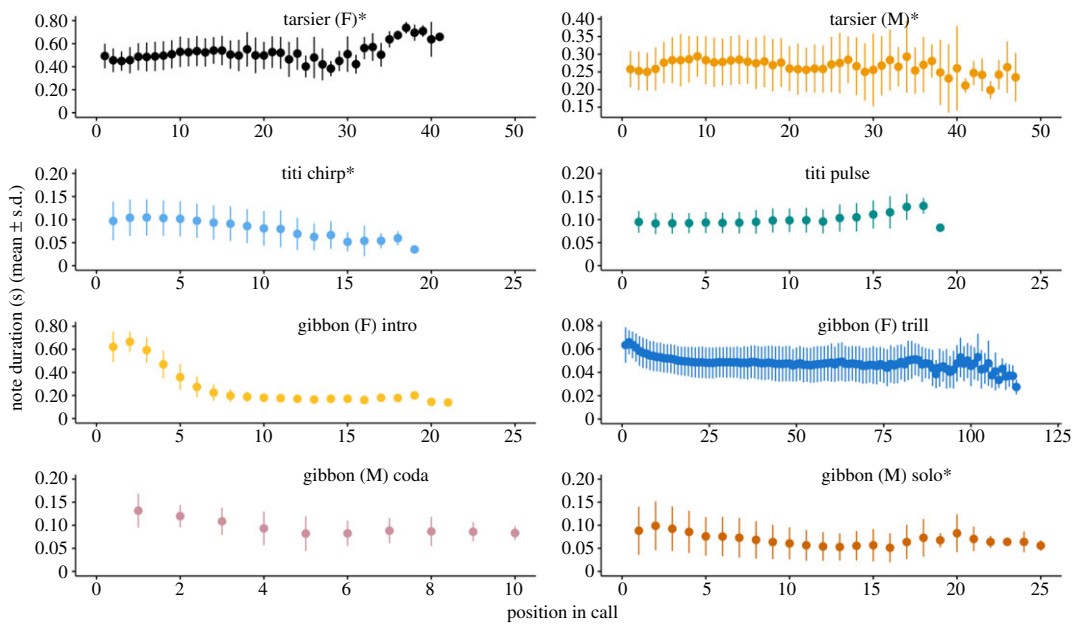

**Figure 5.** Relationship between note duration in seconds and position in call sequence for all call types examined. The points indicate the mean note duration ± s.d. An asterisk next to the call type indicates that particular call type adhered to Menzerath's Law.

coefficient estimates ±95% confidence intervals of GLMMS for all call types included in the analysis. Table 2 also includes pseudo-$R^2$ (fixed), which indicates the amount of variance explained by model predictors, and pseudo-$R^2$ (total), which indicates the variance explained by the full model.

For titi monkey pulses, the top model included the number of notes as a positive predictor of note duration, which is a pattern opposite of that predicted by Menzerath's Law. For gibbon female calls, position in the call sequence was an important predictor for both the introduction and trill (table 2 and figure 5). The number of notes also had a slightly positive effect on note duration for gibbon female introductions (table 2). For male codas, the null model was the highest-ranked model, indicating that neither number of notes nor position in call sequence were reliable predictors (tables 2 and 3). For all call types (except the male coda) the top model performed substantially better than the intercept only model (100% of model weight; $\Delta$AICc > 99; table 3).

# 4. Discussion

The results of our analysis of Menzerath's Law in the duet contributions of non-human primates shows that adherence to the law is not inevitable [34], at least in the specific call types that we examined. Contrary to our predictions, only four of the eight call types examined exhibited patterns consistent with Menzerath's Law. Our investigation revealed two scenarios in which calls did not adhere to Menzerath's Law: when there is a lack of variability in note duration (such as that seen in titi monkey pulses and gibbon female trills) or when the structure of the call leads to a deterministic decrease in note duration over the course of a call (as seen in the gibbon female introductory portion or male gibbon codas). It has been proposed that for long-distance calls, selection pressures for improved transmission are more important than that for efficiency [1,37], which may explain the lack of adherence to Menzerath's Law in our dataset.

Understanding trade-offs in vocal production can provide important insights into the relationship between structure and function of vocal signals, and trade-offs consistent with Menzerath's Law have been attributed to biomechanical constraints on vocal production [32]. An important caveat for the interpretation of trade-offs in vocal production is that in cases where data do not show evidence for trade-offs it does not necessarily indicate that the production constraint is not there, but rather that animals are not pushed to their physiological capacities [65]. We posit that the exceptions are more interesting than the rules in the context of vocal production trade-offs. For example, in both tarsier female duet contributions and titi monkey pulses, note duration increases over the course of a call, which is the opposite of what we predicted if there were energetic or breathing constraints on vocal

**Table 2.** Results of GLMM used to test for the relationship between note duration, number of notes and position in call. Coefficient estimates and 95% confidence intervals are presented for the intercept, number of notes and position in call sequence for each call type. The intercept indicates the estimated value of the outcome variable (in this case note duration) when predictors are equal to zero. We considered predictors (e.g. number of notes or position in call) reliable if the 95% confidence intervals did not overlap zero. Call types which adhered to Menzerath's Law (tarsier female, tarsier male, titi chirp and male gibbon solo) are indicated with an asterisk; these call types had a negative relationship between number of notes and note duration. $R^2$ (fixed) indicates the amount of variance explained by the predictors or fixed effects in the models, whereas $R^2$ (total) indicates the amount of variance explained by the full model. For gibbon females, a random effect was included for individual and site; for the rest of the call types, only a random effect for individual was included.

| | tarsier (F)* | tarsier (M)* | titi chirp* | titi pulse | gibbon (F) intro | gibbon (F) trill | gibbon (M) coda | gibbon (M) solo* |
|---|---|---|---|---|---|---|---|---|
| (intercept) | 0.527 | 0.293 | 0.166 | 0.069 | 0.663 | 0.063 | 0.156 | 0.192 |
| | [0.484, 0.570] | [0.275, 0.311] | [0.153, 0.179] | [0.063, 0.076] | [0.628, 0.698] | [0.059, 0.067] | [0.137, 0.175] | [0.165, 0.219] |
| number of notes | −0.005 | −0.001 | −0.005 | 0.003 | 0.025 | 0.000 | −0.003 | −0.004 |
| | [−0.006, −0.003] | [−0.002, −0.001] | [−0.006, −0.004] | [0.002, 0.003] | [0.021, 0.028] | [0.000, 0.000] | [−0.007, 0.000] | [−0.004, −0.003] |
| position in call | 0.009 | 0.001 | −0.003 | 0.000 | −0.091 | −0.000 | −0.002 | −0.004 |
| | [0.008, 0.011] | [0.001, 0.001] | [−0.003, −0.002] | [0.000, 0.000] | [−0.092, −0.090] | [−0.000, −0.000] | [−0.004, 0.000] | [−0.004, −0.003] |
| $R^2$ (fixed) | 0.091 | 0.038 | 0.080 | 0.066 | 0.669 | 0.074 | 0.011 | 0.073 |
| $R^2$ (total) | 0.231 | 0.228 | 0.406 | 0.655 | 0.723 | 0.170 | 0.269 | 0.356 |

**Table 3.** Akaike's information criterion (AICc) model comparison results. For each call type, we compared four models with note duration as the outcome: the null (or intercept only) model, a model with number of notes as a predictor, a model with the position in call as a predictor and a model with both number of notes and position in the call as predictors of note duration. The top model as indicated by AICc comparison for each call type is shown in bold. For all call types except the male coda the top model performed substantially better than the intercept only model (100% of model weight; $\Delta$AICc > 99).

| models (note duration ~) | AICc | $\Delta$AICc | d.f. | weight |
|---|---|---|---|---|
| tarsier (F) | | | | |
| **~number of notes + position in call** | **−1101.15** | **0.00** | **5** | **1.00** |
| ~position in call | −1045.60 | 55.55 | 4 | 0.00 |
| (intercept) | −954.58 | 146.57 | 3 | 0.00 |
| ~number of notes | −952.40 | 148.75 | 4 | 0.00 |
| tarsier (M) | | | | |
| **~number of notes + position in call** | **−4196.54** | **0.00** | **5** | **1.00** |
| ~number of notes | −4180.11 | 16.44 | 4 | 0.00 |
| (intercept) | −4097.99 | 98.56 | 3 | 0.00 |
| ~position in call | −4082.86 | 113.68 | 4 | 0.00 |
| titi chirp | | | | |
| **~number of notes + position in call** | **−18753.56** | **0.00** | **5** | **1.00** |
| ~number of notes | −18679.56 | 74.00 | 4 | 0.00 |
| ~position in call | −18546.46 | 207.11 | 4 | 0.00 |
| (intercept) | −18289.21 | 464.36 | 3 | 0.00 |
| titi pulse | | | | |
| **~number of notes** | **−17675.92** | **0.00** | **4** | **1.00** |
| ~number of notes + position in call | −17662.96 | 12.96 | 5 | 0.00 |
| ~position in call | −17511.05 | 164.86 | 4 | 0.00 |
| (intercept) | −17494.61 | 181.31 | 3 | 0.00 |
| gibbon (F) intro | | | | |
| **~number of notes + position in call** | **−8028.76** | **0.00** | **6** | **1.00** |
| ~position in call | −7829.64 | 199.12 | 5 | 0.00 |
| ~number of notes | 2682.36 | 10711.12 | 5 | 0.00 |
| (intercept) | 2724.68 | 10753.44 | 4 | 0.00 |
| gibbon (F) trill | | | | |
| **~position in call** | **−334025.72** | **0.00** | **5** | **1.00** |
| ~number of notes + position in call | −334009.89 | 15.83 | 6 | 0.00 |
| ~number of notes | −328877.27 | 5148.45 | 5 | 0.00 |
| (intercept) | −328793.87 | 5231.85 | 4 | 0.00 |
| gibbon (M) coda | | | | |
| **(intercept)** | **−4644.98** | **0.00** | **3** | **0.65** |
| ~number of notes | −4643.43 | 1.56 | 4 | 0.30 |
| ~position in call | −4639.85 | 5.13 | 4 | 0.05 |
| ~number of notes + position in call | −4633.29 | 11.69 | 5 | 0.00 |
| gibbon (M) solo | | | | |
| **~number of notes + position in call** | **−32519.56** | **0.00** | **5** | **1.00** |
| ~number of notes | −32283.03 | 236.54 | 4 | 0.00 |
| ~position in call | −32180.59 | 338.97 | 4 | 0.00 |
| (intercept) | −31212.97 | 1306.59 | 3 | 0.00 |

production [31]. Although we did not include note type in our analysis, another way that animals can deal with breathing constraints is to increase the number of short notes or call types in the vocal sequence; evidence for this was seen in gelada [31] and penguin [19] vocal sequences. In tarsier females, the species-specific structure of the duet contribution results in longer notes at the end of the phrase, and it appears that observed patterns consistent with Menzerath's Law result from adding additional shorter notes at the start of the phrase. For titi monkey pulses, duetting partners were shown to converge in pulse rate [51], and it may be that selection for vocal plasticity in duet pulses was stronger than selection for efficiency.

Comparing different call types within species and/or individuals (such as the introductory versus trill portion of the female gibbon call, or the male gibbon coda and solo) can help us understand potential differences in the investment of different call types [32]. Gibbon female trills exhibit a trade-off between note rate and note bandwidth, such that faster trills tended to have lower bandwidth [52], and there is a negative relationship between introduction duration and trill duration [47]. We found that trill notes tend to decrease in duration over the course of a trill, but there was not a negative relationship between the number of trill notes and trill note duration. The lack of variability in trill note duration, along with the trade-off in trill rate and note bandwidth, indicate that constraints other than those associated with Menzerath's Law have shaped gibbon trills. In male gibbon solos, duration decreased over the course of the call, indicating that breathing constraints may have shaped these calls. Whereas, our top model for male codas did not include either of our predictor variables. Male codas are emitted after the female call, and it has been shown that males flexibly time their codas in relation to the female contribution [66], and that codas increase in complexity over the course of a duet [67]. Therefore, selection pressures (apart from those that lead to patterns consistent with Menzerath's Law) may have been important in shaping the organization of male codas. Importantly, and contrary to our predictions that duet contributions across taxa would be subject to similar selection pressures, it appears that different selection pressures have shaped the duet contributions of male and female primates across taxa.

There are a few caveats that limit our ability to draw overarching conclusions related to compression in vocal communication systems of duetting primates. First, our analyses focused on calls emitted during a single context: the duet. Although testing for Menzerath's Law in primate duets provided a useful comparative framework, future studies that investigate adherence to Menzerath's Law across the full vocal repertoire of each species will help improve our understanding of the generalizability of this law. Second, we followed the general convention for studies of primate duets and focused on specific call types (e.g. titi monkey chirps versus pulses) within the duet bouts. It is possible that if instead of focusing our analysis on specific call types within the duet, we conducted a different analysis using all of the notes in the entire duet bout (*sensu* [20]) then our results would be different. Although not impossible, an analysis such as this would be much less straightforward with our data, particularly in cases of titi monkey and Bornean gibbon duets where it is often difficult to assign each individual note in a duet to the individual caller. Third, given the structure of our dataset, we were not able to test for adherence to Zipf's Law of Brevity, as an effective test of this law would require calls to contain distinct note types [8,68]. Apart from the gibbon female calls which contain a series of long introductory notes that transition into shorter trill notes, the notes in all other call types examined were highly intergraded and preclude easy classification by a human observer. There has been mixed support for Zipf's Law of Brevity in gibbon male solos from different species [20,21], and future analyses of entire duet bouts that distinguish between distinct note types will be informative.

# 5. Conclusion

Despite the prevalence of Menzerath's Law across diverse biological systems [69] and previous reports documenting patterns consistent with the law in chimpanzee [32] and male gibbon [20,21] vocalizations, we found mixed support for Menzerath's Law in primate duets. As adherence to Menzerath's Law is a prediction of compression, it may be that selection pressures related to vocal plasticity or biomechanical constraints on note production have been more important in shaping primate duets than selection for compression or efficiency. We show that for Menzerath's Law to be applicable to the vocal system, there must be sufficient variation in note duration, and call structure cannot have a deterministic decrease in note duration over the course of a call. Identification of universals in vocal communication is a worthwhile goal that can help illuminate the fundamental principles that shape signal evolution [1,31,32], but future work is needed before making strong

conclusions about the universality and broad-scale applicability of Menzerath's Law, and subsequently compression, in primate communication systems.

Ethics. All data presented are from previously published work.

Data accessibility. Data and R code needed to recreate analyses and figures are available via https://dx.doi.org/10.5061/dryad.dz08kprw6 [70].

Authors' contributions. Both authors contributed equally to data collection, analysis and writing of the manuscript.

Competing interests. We declare we have no competing interests.

Funding. We received no funding for this study.

Acknowledgements. We acknowledge Abdul Hamid Ahmad and Johny S. Tasirin for acting as our local collaborators in Malaysia and Indonesia, respectively. We thank all of the interns at UC Davis that worked with D.J.C. and A.R.L. to annotate call spectrograms. We thank Dr Karen Bales for her feedback on an earlier version of this manuscript. We gratefully acknowledge our funding sources for the titi monkeys included in this analysis, the National Institutes of Health grant nos. HD092055 and OD011107. We acknowledge the Fulbright programme for providing funding for the collection of field data in Malaysia and Indonesia.

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
