## [Reviewer comments · Royal Society Open Science]

Review History

RSOS-201557.R0 (Original submission)

Review form: Reviewer 1

Is the manuscript scientifically sound in its present form?

Yes

Are the interpretations and conclusions justified by the results?

Yes

Is the language acceptable?

Yes

Do you have any ethical concerns with this paper?

No

Have you any concerns about statistical analyses in this paper?

No

Recommendation?

Accept with minor revision (please list in comments)

Comments to the Author(s)

The authors made several changes to enhance their manuscript, and I satisfied that these changes address the reviewers' feedback. Thank you to the authors for their hard work – I'm excited for this paper to be published and inspire further discussions about efficiency in animal communication. I only have suggestions for minor revisions.

- Lines 30-31: I am a bit confused by the phrasing "pressures for efficient long-range communication of duets was stronger than that of compression". Efficiency and compression seem like analogous concepts. Do the authors mean to say that long-range signals may need to be conspicuous rather than efficient/compressed?

- Line 142: Please include version information for Raven Pro.

- Line 166 or 193-195: Please include version information for R.

- Figures 2-5, Tables 1-3: Can the authors include species information for "male coda" and "male solo"? I realize that this information is in the main text, but it would be helpful to also have this information available in the figures and/or legends. It is not obvious that male coda and male solo are from gibbons unless a reader looks for this information in the main text.

- Figure 2: Can the authors move the subplot ids (A, B) to the top left of each plot? It is a bit confusing to have the labels on the right-hand side.

- Figures 3,5: The legends say that the points are mean note duration and SEM, but I cannot see the error bars in most of the plots. There are a few things the authors can do: Make the plots longer (rather than wide) so that the points/line stretch out vertically, make the points smaller so that the error lines are not covered up, and use SD rather than SEM.

- Figures, 3,5: I have an aesthetic suggestion: can authors line up y-axes across columns? The figures look a bit messy because the plots are not aligned.

- Figure 4: Can the authors show all call types, rather than just the significant models? Asterisks could be used to denote models with significant coefficients... I also have an aesthetic suggestion: can the authors use the same color for each species and different point symbols (pch in R) to denote different call types? This will help readers easily visualize which species/call type is which.

- Figures 2,3,5: I really like the colored symbols in Fig 4 to visualize species. I suggest that the authors consider applying these species colors and call type symbols across all plots. This way, readers can easily see which data refer to which species/call throughout the manuscript. It is not necessary for the authors to do this, but I think this change could enhance the data visualization.

- Line 345: This phrasing is a bit awkward, "...limitations to the present study that limit our...". Change limitations to caveats?

Review form: Reviewer 2

Is the manuscript scientifically sound in its present form?

Yes

Are the interpretations and conclusions justified by the results?

Yes

Is the language acceptable?

Yes

Do you have any ethical concerns with this paper?

No

Have you any concerns about statistical analyses in this paper?

No

Recommendation?

Accept as is

Comments to the Author(s)

Dear Editor,

I have commented on a previous version of the manuscript and I am pleased to see that all my earlier concerns have been carefully addressed.

I would therefore recommend the publication of the paper in Royal Society Open Science.

Review form: Reviewer 3

Is the manuscript scientifically sound in its present form?

Yes

Are the interpretations and conclusions justified by the results?

Yes

Is the language acceptable?

Yes

Do you have any ethical concerns with this paper?

No

Have you any concerns about statistical analyses in this paper?

No

Recommendation?

Accept as is

Comments to the Author(s)

I was Reviewer #1 on the original submission to Biology Letters. I think the authors have done excellent work handling the various comments raised not just by me but also by the other reviewers. The expanded format here allows for a more detailed explanation of issues central to the argument, and the revised manuscript is really good and effective. I have only one very small comment. In the revised introduction, line 52, the term "compression" seems to come out of nowhere, as it hadn't yet been linked to the ideas of Zipfs or Menzerath's laws.

Decision letter (RSOS-201557.R0)

Dear Professor Clink,

On behalf of the Editors, we are pleased to inform you that your Manuscript RSOS-201557 "Adherence to Menzerath's Law is the exception (not the rule) in three duetting primate species" has been accepted for publication in Royal Society Open Science subject to minor revision in accordance with the referees' reports. Please find the referees' comments along with any feedback from the Editors below my signature.

Please submit your revised manuscript and required files (see below) no later than 7 days from today's (ie 19-Oct-2020) date. Note: the ScholarOne system will 'lock' if submission of the revision is attempted 7 or more days after the deadline. If you do not think you will be able to meet this deadline please contact the editorial office immediately.

on behalf of Dr Oliver Schülke (Associate Editor) and Kevin Padian (Subject Editor)
openscience@royalsociety.org

Associate Editor Comments to Author (Dr Oliver Schülke):

Dear Dr. Clink,

as the associate editor handling your submission, I have now received comments from all your original reviewers. I agree with them that you did an excellent job in addressing all comments and revising the manuscript accordingly.

Two reviewers have a few tiny comments left and I suggest you revise the ms accordingly to make it even more efficient in communicating your interesting results.

With kind regards,
Oliver Schülke

Reviewer comments to Author:

Reviewer: 1

Comments to the Author(s)

The authors made several changes to enhance their manuscript, and I satisfied that these changes address the reviewers' feedback. Thank you to the authors for their hard work - I'm excited for this paper to be published and inspire further discussions about efficiency in animal communication. I only have suggestions for minor revisions.

- Lines 30-31: I am a bit confused by the phrasing "pressures for efficient long-range communication of duets was stronger than that of compression". Efficiency and compression seem like analogous concepts. Do the authors mean to say that long-range signals may need to be conspicuous rather than efficient/compressed?

- Line 142: Please include version information for Raven Pro.

- Line 166 or 193-195: Please include version information for R.

- Figures 2-5, Tables 1-3: Can the authors include species information for "male coda" and "male solo"? I realize that this information is in the main text, but it would be helpful to also have this information available in the figures and/or legends. It is not obvious that male coda and male solo are from gibbons unless a reader looks for this information in the main text.

- Figure 2: Can the authors move the subplot ids (A, B) to the top left of each plot? It is a bit confusing to have the labels on the right-hand side.

- Figures 3,5: The legends say that the points are mean note duration and SEM, but I cannot see the error bars in most of the plots. There are a few things the authors can do: Make the plots longer (rather than wide) so that the points/line stretch out vertically, make the points smaller so that the error lines are not covered up, and use SD rather than SEM.

- Figures, 3,5: I have an aesthetic suggestion: can authors line up y-axes across columns? The figures look a bit messy because the plots are not aligned.

- Figure 4: Can the authors show all call types, rather than just the significant models? Asterisks could be used to denote models with significant coefficients... I also have an aesthetic suggestion: can the authors use the same color for each species and different point symbols (pch in R) to denote different call types? This will help readers easily visualize which species/call type is which.

- Figures 2,3,5: I really like the colored symbols in Fig 4 to visualize species. I suggest that the authors consider applying these species colors and call type symbols across all plots. This way, readers can easily see which data refer to which species/call throughout the manuscript. It is not necessary for the authors to do this, but I think this change could enhance the data visualization.

- Line 345: This phrasing is a bit awkward, "...limitations to the present study that limit our...". Change limitations to caveats?

Reviewer: 2
Comments to the Author(s)

Dear Editor,

I have commented on a previous version of the manuscript and I am pleased to see that all my earlier concerns have been carefully addressed.

I would therefore recommend the publication of the paper in Royal Society Open Science.

Reviewer: 3
Comments to the Author(s)

I was Reviewer #1 on the original submission to Biology Letters. I think the authors have done excellent work handling the various comments raised not just by me but also by the other reviewers. The expanded format here allows for a more detailed explanation of issues central to the argument, and the revised manuscript is really good and effective. I have only one very small comment. In the revised introduction, line 52, the term "compression" seems to come out of nowhere, as it hadn't yet been linked to the ideas of Zipfs or Menzerath's laws.

===PREPARING YOUR MANUSCRIPT===

- one version identifying all the changes that have been made (for instance, in coloured highlight, in bold text, or tracked changes);
- a 'clean' version of the new manuscript that incorporates the changes made, but does not highlight them. This version will be used for typesetting.

===PREPARING YOUR REVISION IN SCHOLARONE===

Author's Response to Decision Letter for (RSOS-201557.R0)

See Appendix A.

Decision letter (RSOS-201557.R1)

Dear Professor Clink,

It is a pleasure to accept your manuscript entitled "Adherence to Menzerath's Law is the exception (not the rule) in three duetting primate species" in its current form for publication in Royal Society Open Science.

on behalf of Dr Oliver Schülke (Associate Editor) and Kevin Padian (Subject Editor)
openscience@royalsociety.org

Appendix A

Reviewer comments to Author:

Reviewer: 1

Comments to the Author(s)

The authors made several changes to enhance their manuscript, and I satisfied that these changes address the reviewers' feedback. Thank you to the authors for their hard work – I'm excited for this paper to be published and inspire further discussions about efficiency in animal communication. I only have suggestions for minor revisions.

- Lines 30-31: I am a bit confused by the phrasing "pressures for efficient long-range communication of duets was stronger than that of compression". Efficiency and compression seem like analogous concepts. Do the authors mean to say that long-range signals may need to be conspicuous rather than efficient/compressed?

We modified the line slightly to clarify:

It is possible that selection pressures for long-range signals that can travel effectively over large distances was stronger than that of compression in primate duets.

- Line 142: Please include version information for Raven Pro.

We added the relevant information.

For all calls included in the present analysis, individual notes and calls were annotated by human observers using selection tables in Raven Pro version 1.5 or 1.6 (Cornell Lab of Ornithology Center for Conservation Bioacoustics, Ithaca, NY, U.S.A.).

- Line 166 or 193-195: Please include version information for R.

We added the following information:

To test for adherence to Menzerath's law in each call type, we created four generalized linear mixed models (GLMMs) using the R package 'lme4' [58] in the R programming environment version 3.6.2 [59].

- Figures 2-5, Tables 1-3: Can the authors include species information for "male coda" and "male solo"? I realize that this information is in the main text, but it would be helpful to also have this information available in the figures and/or legends. It is not obvious that male coda and male solo are from gibbons unless a reader looks for this information in the main text.

Thank you for this suggestion. We updated the figures and tables throughout the manuscript.

- Figure 2: Can the authors move the subplot ids (A, B) to the top left of each plot? It is a bit confusing to have the labels on the right-hand side.

Based on the previous comment we modified the axis labels to indicate that the male gibbon call types came from male gibbons. This led to the x-axes looking a bit crowded with the previous layout, so we now have the two plots in a single column as opposed to side by side. We did not move the subplot IDs (as it was difficult to read with the labels on the left side) but

we feel that the new layout is much more clear.

- Figures 3,5: The legends say that the points are mean note duration and SEM, but I cannot see the error bars in most of the plots. There are a few things the authors can do: Make the plots longer (rather than wide) so that the points/line stretch out vertically, make the points smaller so that the error lines are not covered up, and use SD rather than SEM.

We followed the reviewer suggestions and feel that the new versions of the plots are much improved.

- Figures, 3,5: I have an aesthetic suggestion: can authors line up y-axes across columns? The figures look a bit messy because the plots are not aligned.

Agreed and we modified both the x and y axes so that they plots look cleaner.

- Figure 4: Can the authors show all call types, rather than just the significant models? Asterisks could be used to denote models with significant coefficients... I also have an aesthetic suggestion: can the authors use the same color for each species and different point symbols (pch in R) to denote different call types? This will help readers easily visualize which species/call type is which.

We added all call types (except male coda as the null model was ranked highest). We tried to change the shapes so that they correspond with the call types, but since the 90% confidence intervals are small it was even harder to see with different shapes. Therefore we opted to use only colors but not shapes to distinguish between call types.

- Figures 2,3,5: I really like the colored symbols in Fig 4 to visualize species. I suggest that the authors consider applying these species colors and call type symbols across all plots. This way, readers can easily see which data refer to which species/call throughout the manuscript. It is not necessary for the authors to do this, but I think this change could enhance the data visualization.

We really liked this idea and modified the plots accordingly.

- Line 345: This phrasing is a bit awkward, "...limitations to the present study that limit our...". Change limitations to caveats?

We changed the line to the following:

There are a few caveats that limit our ability to draw overarching conclusions related to compression in vocal communication systems of duetting primates.

Reviewer: 2

Comments to the Author(s)

Dear Editor,

I have commented on a previous version of the manuscript and I am pleased to see that all my

earlier concerns have been carefully addressed.

I would therefore recommend the publication of the paper in Royal Society Open Science.

Reviewer: 3

Comments to the Author(s)

I was Reviewer #1 on the original submission to Biology Letters. I think the authors have done excellent work handling the various comments raised not just by me but also by the other reviewers. The expanded format here allows for a more detailed explanation of issues central to the argument, and the revised manuscript is really good and effective. I have only one very small comment. In the revised introduction, line 52, the term "compression" seems to come out of nowhere, as it hadn't yet been linked to the ideas of Zipfs or Menzerath's laws.

Thank you for catching this. We modified the line as outlined below:

There has been increasing interest in applying these statistical laws outside of human language [10,11,20,21,12–19]. **These statistical laws reflect compression**, and it has been proposed that compression is a universal principle not just in vocal communication but in behavior more broadly [1].